# Cocaine Abuse as an Immunological Trigger in a Case Diagnosed with Eales Disease

**DOI:** 10.3390/medicina59010169

**Published:** 2023-01-14

**Authors:** Ludovico Iannetti, Fabio Scarinci, Ludovico Alisi, Alessandro Beccia, Andrea Cacciamani, Maria Carmela Saturno, Magda Gharbiya

**Affiliations:** 1Department of Sense Organs, Sapienza University of Rome, Head and Neck Department, Policlinico Umberto I University Hospital, Viale del Policlinico 155, 00161 Rome, Italy; 2IRCCS Fondazione Bietti, 00198 Rome, Italy; 3Clinica Mediterranea, 80122 Naples, Italy

**Keywords:** Eales disease, cocaine, factor V Leiden mutation, drug abuse, trombophilic mutation

## Abstract

*Background*: Eales disease is a clinical syndrome affecting the mid-peripheral retina with an idiopathic occlusive vasculitis and possible subsequent retinal neovascularization. The disease can develop into visually threatening complications. *Case Presentation*: We report the case of a 40-year-old Caucasian male with a history of cocaine abuse who presented with blurred vision in the left eye (LE). Fundus examination showed vitreous hemorrhages, peripheral sheathing of venous blood vessels, areas of retinal neovascularization in the LE, and peripheral occlusive phlebitis in the right eye. The full serologic panel was negative except for the heterozygous mutation of factor V Leiden. Clinical and biochemical parameters suggested a diagnosis of Eales disease. Therapy with dexamethasone, 1 mg per kg per day, tapered down slowly over 4 months, and peripheral laser photocoagulation allowed a regression of clinical signs and symptoms. *Conclusion*: This case shows an uncommon presentation of Eales disease associated with cocaine abuse. Both cocaine abuse and a thrombophilic pattern, as cofactors, might have sensitized the retinal microcirculation on the pathogenetic route to this retinal pathology. Furthermore, in view of this hypothesis, a thorough ocular and general medical history investigating drug abuse and coagulation disorders is recommended for ophthalmologists in such cases.

## 1. Introduction

Eales disease is a clinical syndrome with four evolutionary stages: the first stage, also known as the inflammatory stage, is characterized by perivasculitis affecting electively the peripheral retina. The second stage, or the ischemic stage, shows peripheral retinal capillary non-perfusion alongside retinal vein vessel changes (sclerosis, irregularity of vein caliber, and abnormal vascular anastomosis). In the third stage, the patient develops fibrovascular proliferation and vitreous hemorrhage. The last stage is characterized by a mixed tractional-rhegmatogenous retinal detachment, neovascularization of the iris, neovascular glaucoma, complicated cataract, and optic atrophy [1,2]. The exposition to Mycobacterium Tuberculosis has been suggested to play a pivotal role as the primary etiology for this disease [3]. The treatment is based on oral corticosteroids and laser photocoagulation in the early phases, and vitrectomy in more advanced stages [1,2].

In this case, we hypothesize that cocaine as an exogenous agent might be the immunological trigger for a predisposed patient to develop Eales disease.

## 2. Case Presentation

A 40-year-old Caucasian male presented complaining of blurred vision and floaters in the left eye (LE) for the last two months. At presentation, the patient reported a 10-year history of chronic cocaine abuse and a family history of cerebral strokes at a young age (his mother), and he was a heavy smoker (more than 20 cigarettes per day). The patient was systemically healthy, and there was no apparent medical history contributing to the retinal problem. The patient presented with best-corrected visual acuity (BCVA) of 20/20 in the right eye (RE) and 20/30 in the left eye (LE). No evidence of iris neovascularization, cells, or flare in the anterior chamber was noticed in both eyes (BE), whereas a fresh vitreous hemorrhage was observed in the LE. The intraocular pressure was within the limit in BE (16 mmHg in the RE and 14 mmHg in the LE).

The fundus examination showed signs of peripheral occlusive phlebitis in the BE (Figure 1A1) and partial vitreous hemorrhage with areas of retinal neovascularization in the LE (Figure 1B1). A wide-field fundus fluorescein angiography (FFA) showed delayed venous filling, staining of inflamed peripheral vessel walls in BE, and areas of retinal capillary nonperfusion with 360-degree peripheral ischemic areas in BE (Figure 1A2,B2).

Macular spectral optical coherence tomography was normal in BE. Among the required tests, quantiFERON TB-gold, the venereal disease reference laboratory, and the Treponema pallidum hemagglutination assay were negative, as were the autoantibodies (antinuclear antibodies, extractable nuclear antigen antibodies, anti-neutrophil cytoplasmic antibodies, and anti-cardiolipin antibodies). The CT scan was negative, and the ACE test was within normal ranges. The clinical features were consistent with a diagnosis of Eales disease.

Given the family history of thrombotic phenomena, further examination, including inherited disorders of coagulation, was obtained. This showed the presence of the heterozygous form of the factor V Leiden mutation, while the rest of the coagulation panel and homocysteine were normal. Therefore, therapy with oral dexamethasone (1 mg per kg per day, to taper down slowly in 4 months) was administered. Peripheral laser photocoagulation of the ischemic areas in BE was performed, and an FFA was repeated 2 months after laser treatment. (Figure 2) After a 10-month follow-up, the steroid treatment was discontinued, and the BCVA was 20/20 in BE. The vitreous hemorrhage in the LE had completely resolved, and no area of retinal neovascularization was detected at the FFA in the BE. Written consent to publish the case was obtained from the patient.

## 3. Discussion and Conclusions

Eales disease is presumed to be a consequence of an immunological reaction in response to exogenous agents. Low levels of antioxidants such as glutathione, alteration of the C3 complement cascade, and HLA arrays such as B5, DR1, and DR4 have been recently reported to be associated with Eales disease [1], while the association with Mycobacterium Tuberculosis is widely described [2].

In the context of vascular occlusive diseases, in which endothelial toxicity and thrombophilia are the main pathological events, the concurrent history of factor V Leiden mutation and cocaine drug abuse, in our case, is noteworthy. In fact, we hypothesize that cocaine, as an exogenous agent, might be the immunological trigger in a predisposed patient with heterozygosis of the factor V Leiden mutation to develop Eales disease.

While the association between retinal vein occlusion and congenital thrombophilic diseases such as anticardiolipin antibodies, hyperhomocysteinemia, and the factor V Leiden mutation is a widely recognized clinical entity [4], only one case of peripheral retinal vasculitis associated with the heterozygosity factor V Leiden mutation was described [5].

In this regard, heterozygosity for the factor V Leiden mutation was associated with a sevenfold augmented risk of thrombosis, while homozygosity was correlated with an increased thrombotic risk up to 79 times in comparison with the healthy subjects.

Cocaine is well-known for its effects on the cardiovascular system, such as vasoconstriction, endothelial vessel dysfunction, and hypertension, which may lead to a prothrombotic state [6]. In addition, some authors have pointed out the retinal effects of cocaine abuse on the hypoperfusion of the deep capillary plexus [7] and the bilateral enlargement of the foveal avascular zone [8]. Moreover, the local adverse effects of cocaine include ischaemic optic neuropathy, central retinal artery, and vein occlusions. Cocaine is oftentimes contaminated with talc or other diluting substances that may determine retinopathy from contaminants. A recent report from Rahman et al. suggested that cocaine may be able to induce severe vasospasm and hypertension that may ultimately lead to chorioretinal ischemia [9].

Although Eales disease is a primary venous pathology and cocaine effects on the venous vessels are poorly understood, in our case, the resulting hypoxia due to cocaine abuse associated with the thrombophilic status could represent a predisposing condition or “the straw that breaks the camel’s back” in the development of peripheral vascular occlusive disease. In a similar case, Roohipourmoallai et al. reported the development of vitreous hemorrhage in a cocaine user due to peripheral neovascularization and peripheral avascular retina. The authors suggest that such a condition may have been determined by the inhalation of talc particles in the context of talc retinopathy [10].

Furthermore, as with drugs such as cocaine, nicotine has been shown to activate the reward circuits of the brain and increase levels of the chemical messenger dopamine, which might reinforce, in such cases, rewarding behaviors [11].

In the present case, both oral corticosteroids and laser treatment were performed. Oral dexamethasone has shown considerable efficacy in Eales disease, especially in the reduction of the inflammatory component that characterized the acute stages. Biswas et al. reported significantly better visual outcomes at the final follow-up compared to patients who did not receive oral steroids [12]. Likewise, in the occlusive and neovascular stages of the disease, peripheral laser photocoagulation has demonstrated better visual outcomes during follow-up. Therefore, the available literature strongly suggests an aggressive treatment for Eales disease to prevent the development of later stages [12,13].

Although this is a rare eventuality, in the present clinical case we could speculate that both cocaine abuse and a thrombophilic pattern, as cofactors, might have sensitized the retinal microcirculation on the pathogenetic route to Eales disease. Additionally, in view of this hypothesis, a thorough ocular and general medical history that includes searching for drug abuse and coagulation disorders is recommended for ophthalmologists dealing with Eales disease. However, given the rarity of both the factor V Leiden mutation and Eales disease, further investigations are needed to support our conclusions.

## Figures and Tables

**Figure 1 medicina-59-00169-f001:**
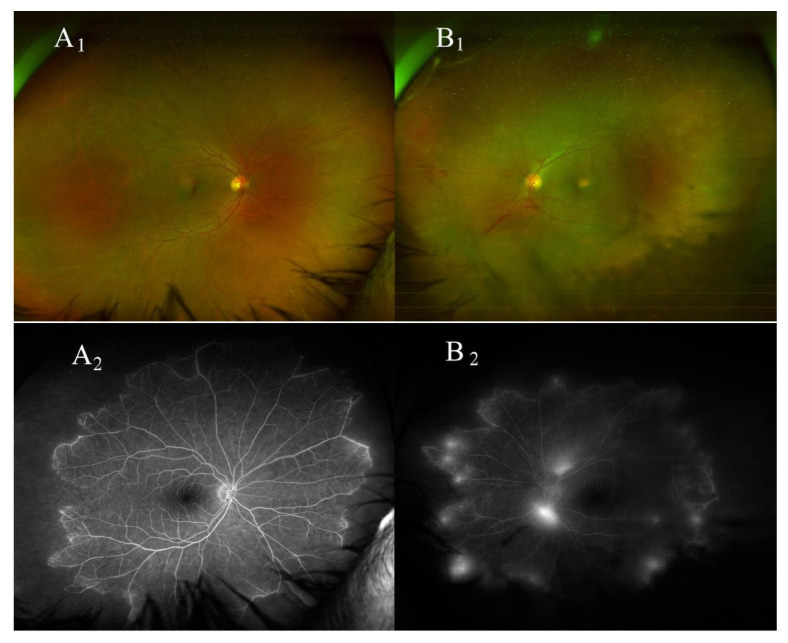
In the right eye, fundus imaging (**A1**) shows peripheral periphlebitis, and wide field fundus fluorescein angiography (**A2**) shows staining of peripheral vessels as well as areas of retinal capillary nonperfusion with 360-degree peripheral ischemic areas. In the left eye, fundus imaging (**B1**) shows vitreous hemorrhages as well as retinal neovascularization at the disk area, and in the peripheral retina, wide field fundus fluorescein angiography (**B2**) shows areas of retinal capillary nonperfusion with 360-degree peripheral ischemic areas as well as areas of leakage from retinal neovascularization.

**Figure 2 medicina-59-00169-f002:**
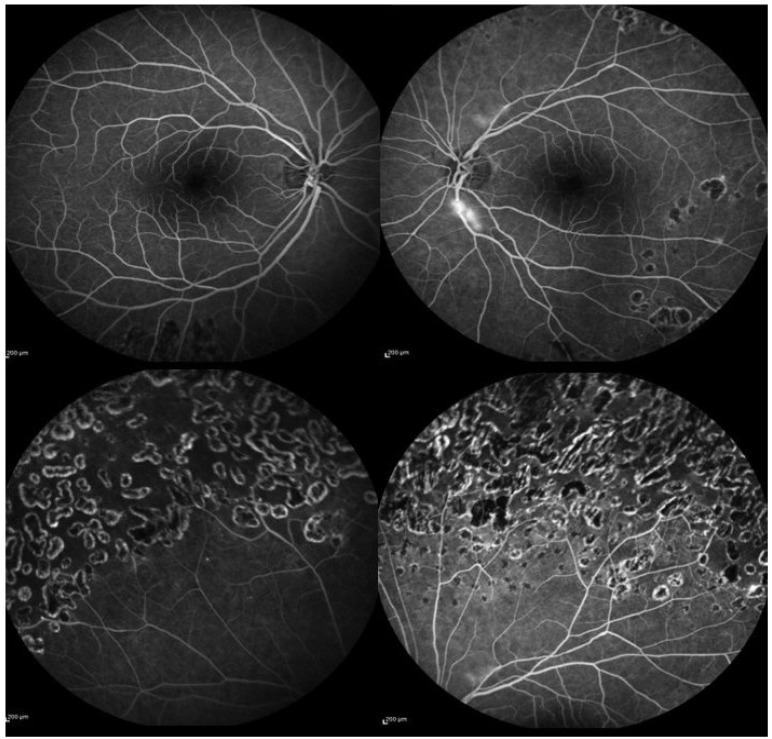
Fundus fluorescein angiography showing regression of retinal neovascularization in the left eye (**upper right**). Normal fundus appearance for the posterior pole in the right eye (**upper left**). Peripheral laser scars in both eyes (**lower left** and **right**).

## Data Availability

Data are contained within the article.

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
