# Peer review of "Cocaine Abuse as an Immunological Trigger in a Case Diagnosed with Eales Disease"

_medicina, 2023, doi:10.3390/medicina59010169_

Round 1
Reviewer 1 Report
1. Please provide the fundus photographas after laser treatment
2. Please describe more precisely the rationale behind steroid treatment in that particular patient.
3. Please elaborate more on the possible link between cocaine use and Eales onset, occurrence.
Author Response
Thank you for your interesting comments and suggestions.
- Please provide the fundus photographs after laser treatment.
A: Unfortunately, we can’t provide the fundus image after laser treatment. We added the FAF after laser treatment to the case report.
- Please describe more precisely the rationale behind steroid treatment in that particular patient.
A: We added a paragraph in the discussion to explain more in detail the rationale behind our choice of treatment. (lines 131-138).
- Please elaborate more on the possible link between cocaine use and Eales onset, occurrence.
A: We expanded the discussion (lines 114-119 and 124-127).
Reviewer 2 Report
A case report based on Eales disease (Cocain abuse as an immunological trigger) is reported in this work. An author should refine the manuscript on the following points:
1. There is a need to provide additional information regarding the results included.
2. There is a need to improve the English language throughout.
3. Pls comment on "non-perfusion" in BE.
4. There must be proper and recent citations.
5. The Figure 1 needs to be describe well .
6. There needs to be improved in the abstract and result & discussion part.
7. Pls provide more details on sentence (line 60) "Intraocular pressure was within the limit in BE"
Author Response
We’d like to thank the reviewer for the punctual observations and precise indications.
- There is a need to provide additional information regarding the results included.
A: Amended
- There is a need to improve the English language throughout.
A: We improved our article with a native English speaker.
- Pls comment on "non-perfusion" in BE.
A: Fundus fluorescein angiography showed areas of retinal capillary nonperfusion in both eyes, i.e. retina did not have efficient tissue perfusion, showing consequently 360 degrees peripheral ischemic areas in both eyes (lines 68-69).
- There must be proper and recent citations.
A: We updated and expanded the reference section.
- The Figure 1 needs to be describe well.
A: We have better described Figure 1 in the text and in the legend.
- There needs to be improved in the abstract and result & discussion part.
A: We expanded the discussion section as requested (Lines 114-119, 124-127, and 131-138).
- Pls provide more details on sentence (line 60) "Intraocular pressure was within the limit in BE"
A: Amended.
Round 2
Reviewer 2 Report
NA